# Genome Sequences of Two *Pseudomonas aeruginosa* Isolates with Defects in Type III Secretion System Gene Expression from a Chronic Ankle Wound Infection

Sardar Karash,[a] Robert Nordell,[a] Egon A. Ozer,[b] (ID) Timothy L. Yahr[a]

[a]Department of Microbiology and Immunology, University of Iowa, Iowa City, Iowa, USA
[b]Department of Medicine, Northwestern Fienberg School of Medicine, Chicago, Illinois, USA

**ABSTRACT** Effector proteins translocated into host cells by the *Pseudomonas aeruginosa* type III secretion system (T3SS) are critical for phagocytic avoidance and systemic spread of the microorganism. The T3SS genes are present in virtually all *P. aeruginosa* strains. When examined in environmental isolates and clinical specimens, expression of the T3SS genes is the rule. Isolates from the airways of cystic fibrosis (CF) patients are one exception, and these isolates usually carry mutations that disable T3SS gene expression. In this study, we describe two *P. aeruginosa* isolates, one pigmented brown and one green, from a keratitis-ichthyosis-deafness (KID) syndrome patient with a chronic cutaneous ankle wound. Similar to most isolates from CF, both of the KID isolates were defective for T3SS gene expression. Providing the primary activator of T3SS transcription (*exsA*) in *trans* restored T3SS function. Since the *exsA* sequences were identical to that of a reference strain with active T3SS gene expression, we examined the cAMP-Vfr system, a critical regulator of T3SS gene expression. Vfr is a cAMP-dependent transcription factor that activates *exsA* expression. Whereas T3SS activity was corrected in the brown isolate by restoring cAMP synthesis, the same was not observed for the green isolate. These findings suggest that distinct mechanisms resulted in loss of T3SS gene expression in the KID isolates. The mutations responsible for the T3SS defects were not clearly evident by comparison of the whole-genome sequences to a reference strain. Our findings suggest that loss of T3SS gene expression may be a trait common to both CF and non-CF chronic infections.

**IMPORTANCE** A common feature of microorganisms that cause chronic infections is a stealthy lifestyle that promotes immune avoidance and host tolerance. During chronic colonization of cystic fibrosis (CF) patients, *Pseudomonas aeruginosa* acquires numerous adaptations that include reduced expression of some factors, such as motility, O antigen, and the T3SS, and increased expression of other traits, such as biofilm formation. In this study, we report loss of T3SS gene expression in non-CF chronic isolates. This finding suggests that loss of the T3SS may be a common and important trait that contributes to persistence and may open avenues to explore the significance further using non-CF chronic infection models.

**KEYWORDS** *Pseudomonas aeruginosa*, cystic fibrosis, chronic, type III secretion, ExsA, Vfr, cAMP

Address correspondence to Timothy L. Yahr, tim-yahr@uiowa.edu.

**P**seudomonas aeruginosa is an opportunistic pathogen of humans. While infections in the immunocompetent are rare, medical conditions including neutropenia, severe burns, mechanical ventilation, diabetes, and cystic fibrosis (CF) are risk factors for *P. aeruginosa* infection (1, 2). The infections seen in CF patients are distinct from most other types of *P. aeruginosa* infections and characterized by chronic colonization

that persists for the life of the individual, in many cases extending to decades. CF infants and children experience recurrent intermittent colonization with *P. aeruginosa* that can be treatable with aggressive antibiotic treatment (3). Later in life, intermittent colonization transitions to permanent, irreversible chronic colonization that is refractory to antibiotic treatment (4). Irreversible colonization coincides with a decline in lung function and phenotypic changes within the colonizing *P. aeruginosa* population. Those changes include growth as a biofilm, mucoidy, and reduced expression of many virulence factors, including the type III secretion system (T3SS) (5, 6). Mucoidy, resulting from overproduction of the polysaccharide alginate, and biofilm formation are thought to confer a selective advantage by enhancing resistance to a variety of killing mechanisms, including antibiotics, neutrophils, and cationic compounds (7, 8). The T3SS is a critical virulence factor used to translocate effector proteins into eukaryotic cells (9). Although T3SS gene expression is critical in the context of acute infection, it is unclear how, or even if, loss of T3SS gene expression in CF promotes persistence.

Keratitis-ichthyosis-deafness (KID) syndrome is a rare genetic disorder characterized by defects in the corneal surface, thickened skin plaques, hearing impairment, and recurrent bacterial infections (10). A chronic cutaneous ankle wound swab from a patient with KID syndrome was submitted for testing without request. The swab contained *Staphylococcus aureus* and two distinct *P. aeruginosa* strains, brown and green in pigmentation, that we designated Minnesota *P. aeruginosa* brown (MNPAB) and green (MNPAG), respectively. Because loss of T3SS gene expression is common in chronic CF isolates (11), we tested whether the chronic MNPAB and MNPAG isolates are also defective. To monitor T3SS gene expression, we integrated a transcriptional reporter ($P_{exsD}$-*lacZ*) into the chromosomal CTX phage attachment site of both strains. The $P_{exsD}$-*lacZ* reporter is controlled by the ExsA transcription factor, is activated when cells are cultured in growth medium containing EGTA to chelate calcium, and serves as a reliable surrogate for monitoring T3SS expression (12). The strains were cultured in tryptic soy broth under non-inducing (− EGTA) or inducing (+ EGTA) conditions for T3SS gene expression. The laboratory strain PA103 served as a control. Whereas strain PA103 demonstrates EGTA-dependent $P_{exsD}$-*lacZ* reporter activity, neither MNPAB nor MNPAG had detectable levels of activity (Fig. 1A and B). To corroborate the $P_{exsD}$-*lacZ* reporter data, culture supernatants were harvested and subjected to Coomassie staining and immunoblot analyses. The Coomassie-stained supernatant fraction from strain PA103 demonstrated high levels of the T3SS effectors ExoU and ExoT, and this was confirmed by ExoU immunoblotting (Fig. 1C and D, lanes 2). Consistent with the $P_{exsD}$-*lacZ* reporter data, ExoU and ExoT expression were undetectable in MNPAB and MNPAG (Fig. 1C and D, lanes 4). T3SS gene expression in CF isolates can usually be restored by expressing *exsA* in *trans* (13), and this was also the case for MNPAB and MNPAG, as evidenced by ExsA-dependent transcriptional reporter activity (Fig. 1A and B) and the presence of ExoU and ExoT in the culture supernatant fraction (Fig. 1C and D, lanes 6). Complementation of the T3SS defect in MNPAB and MNPAG by ExsA demonstrates that the T3SS genes are present and functional. The *exsA* promoter region and coding sequences in MNPAB and MNPAG were identical to those in strain PA103, suggesting that the lack of T3SS gene expression reflects a regulatory defect.

Vfr (a cAMP-dependent transcription factor) and CyaB (an adenylate cyclase) are required for high levels of *exsA* transcription (14, 15). We next asked whether the expression of either *vfr* or *cyaB* in *trans* would restore T3SS gene expression. Whereas *vfr* expression led to small increases in $P_{exsD}$-*lacZ* reporter activity (Fig. 2A and B) and ExsA levels (Fig. 2C and D, lanes 3 and 4) in both MNPAB and MNPAG, the overall effect on T3SS activity was modest, as evidenced by the lack of ExoU and ExoT in the culture supernatant fraction (Fig. 2C and D, lanes 4). CyaB expression had differential effects on restoring T3SS gene expression. In MNPAB, CyaB expression resulted in high levels of $P_{exsD}$-*lacZ* reporter activity, secreted ExoU, and ExsA expression (Fig. 2A and C, lanes 6). In contrast, expression of CyaB in MNPAG resulted in only a small increase in $P_{exsD}$-*lacZ* reporter activity and had no effect on the levels of secreted ExoU or ExsA

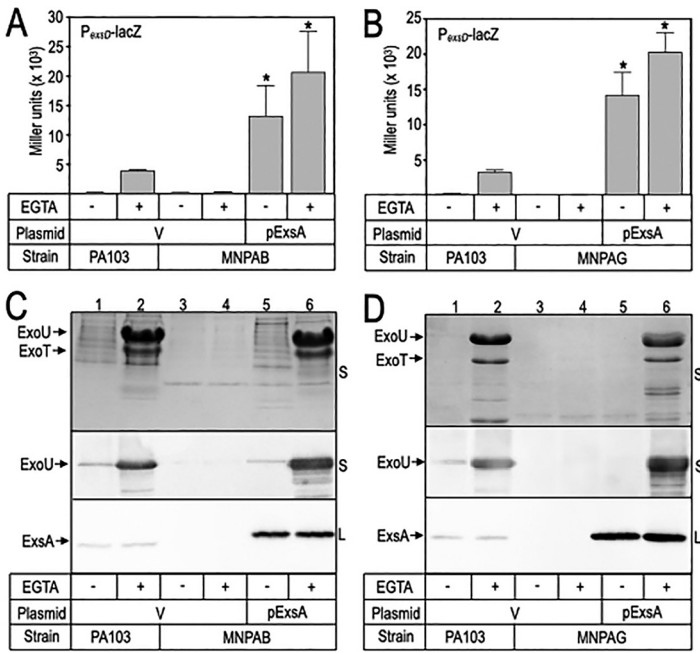

**FIG 1** Type III secretion system expression in *P. aeruginosa* MNPAB and MNPAG. The indicated strains were cultured in tryptic soy broth supplemented with EGTA as indicated to induce T3SS gene expression. Each strain carried either a control plasmid (V) or an ExsA expression plasmid (pExsA). (A, B) Cells were assayed for ExsA-dependent reporter activity ($P_{exsD}$-*lacZ*). The reported values with their standard errors represent the average values from at least three experiments. *, $P < 0.01$. (C, D) Culture supernatant (S) and whole-cell lysate (L) fractions were harvested when the culture $A_{600}$ reached 1.0. The fractions were analyzed by SDS-PAGE and then Coomassie stained (top portion of each panel; ExoU is 70 kDa and ExoT is 53 kDa) or immunoblotted for ExoU or ExsA (32 kDa) as indicated. Laboratory strain PA103 was included as a control.

expression. We conclude that the MNPAB and MNPAG defects in T3SS gene expression are distinct from one another.

To better understand the nature of the T3SS defect, whole-genome sequencing was performed on MNPAB and MNPAG. The MNPAB genome coverage was 43.8×, with an assembled genome size of 6,332,406 bp (66.1% GC) distributed over 151 contigs ($N_{50}$ of 103,810), and the MNPAG genome coverage was 46.12×, with an assembled genome size of 6,683,661 bp (66.1% GC) distributed over 164 contigs ($N_{50}$ of 88,241). MNPAB and MNPAG have 5,885 and 6,207 predicted protein-encoding genes, respectively. *P. aeruginosa* strain PA103 was identified as the most closely related laboratory strain (Fig. S1) and was used as the reference genome for further downstream analyses (16, 17). Although MNPAB and MNPAG were isolated from the same wound and are highly related (Fig. S1 in the supplemental material), the MNPAG genome is ~350 kb larger (Fig. S1). The additional sequences are scattered throughout the assembled genome (Fig. S2) and do not appear to represent plasmids, prophages, or integrative and conjugative elements as assessed with tools that search for these elements (18–21). With respect to virulence, the only notable difference is the absence of the *hcn* gene cluster in MNPAB, which is responsible for hydrogen cyanide production.

Variants that might account for the T3SS defect were identified by comparison of MNPAB and MNPAG to each other and to the PA103 reference strain (summarized in Tables S1 to S3) (22, 23). Similar to our finding with *exsA*, the MNPAB and MNPAG *vfr* and *cyaB* sequences were identical to those in strain PA103, indicating that the T3SS signaling defect occurs upstream of these genes. We chose variants in *fimV* and *bifA* for further analysis, as both could account for the defect in T3SS gene expression. MNPAB contains a frameshift mutation in *fimV*. FimV participates in cAMP synthesis through control of the adenylate cyclase *cyaB* (24). T3SS gene expression was not restored by correcting the *fimV* frameshift mutation on the chromosome (Fig. S3A).

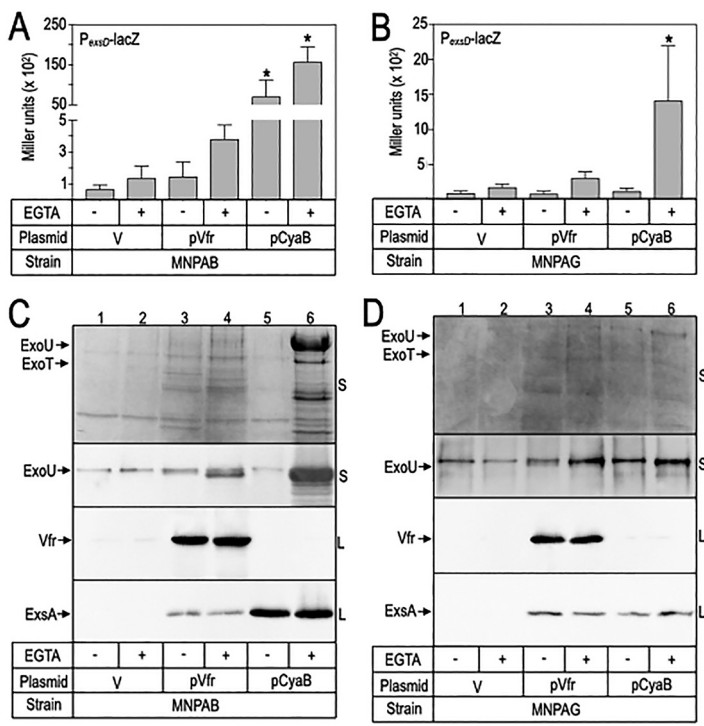

**FIG 2** Restoration of type III secretion system expression in *P. aeruginosa* MNPAB. The indicated strains were cultured in tryptic soy broth supplemented with EGTA as indicated to induce T3SS gene expression. Each strain carried a control plasmid (V), a Vfr expression plasmid (pVfr), or a CyaB expression plasmid (pCyaB). (A, B) Cells were assayed for ExsA-dependent reporter activity ($P_{exsD}$-*lacZ*). The reported values with their standard errors represent the average values from at least three experiments. *, $P < 0.01$. (C, D) Culture supernatant (S) and whole-cell lysate (L) fractions were harvested when the culture $A_{600}$ reached 1.0. The fractions were analyzed by SDS-PAGE and then Coomassie stained (top portion of each panel) or immunoblotted for ExoU, Vfr (24 kDa), or ExsA as indicated.

Another potential explanation was a frameshift mutation in *bifA*, a cyclic-di-GMP phosphodiesterase (25). Elevated levels of cyclic-di-GMP can reduce cAMP levels (26). Expression of *bifA*, however, did not restore T3SS gene expression (Fig. S3B to C).

Our combined findings suggest that an adaptation commonly observed in chronic CF isolates, i.e., loss of T3SS gene expression, may also contribute to non-CF persistent infections. The nature of the T3SS defects, however, was not evident from the genome sequencing owing to the large number of polymorphisms between MNPAB, MNPAG, and strain PA103. We conclude that the T3SS defect is likely multifactorial in nature.

## MATERIALS AND METHODS

**Bacterial strains, culture conditions, and sample preparation.** *P. aeruginosa* strains were maintained on Vogel-Bonner minimal (VBM) medium. The $P_{exsD}$-*lacZ* transcriptional reporter was integrated at the *attB* site of MNPAB and MNPAG as previously described (12). Reporter activity was measured by growing cells overnight at 37°C in LB. The following day, cultures were diluted to an $A_{600}$ of 0.1 in Trypticase soy broth with EGTA (2 mM) as indicated. Cultures were incubated at 37°C with shaking, and samples were harvested when the culture $A_{600}$ reached 1.0. β-Galactosidase activity was determined with the substrate *ortho*-nitrophenyl-galactopyranoside as previously described (12). The complementation experiments were performed with previously described expression vectors for *exsA*, *vfr*, and *cyaB* (15, 27, 28). Each gene is under the transcriptional control of the arabinose-inducible pBAD promoter, and expression was induced by including 0.2% arabinose in the growth medium. Statistical significance was determined by one-way analysis of variance (ANOVA) using GraphPad Prism version 5.0c for Mac OS X (GraphPad Software, La Jolla, CA). Samples for immunoblots were prepared from the cell-associated (lysate) and trichloroacetic acid-precipitated supernatant fractions as previously described, using rabbit polyclonal antiserum to ExoU, ExsA, and Vfr (12).

**Whole-genome sequencing.** Single colonies of *P. aeruginosa* MNPAB and MNPAG were cultured overnight in LB broth at 37°C. Genomic DNA was extracted using a DNeasy blood and tissue kit (Qiagen) and submitted to the University of Pittsburgh Microbial Genomic Sequencing Center (MiGS) for library preparation using a modified Illumina Nextera XT kit as previously described (29) and sequencing on the

Illumina NextSeq 550 platform with 150-bp paired-end reads. Demultiplexing resulted in 3,208,850 and 3,612,542 sequence reads for *P. aeruginosa* MNPAB and MNPAG, respectively. Trimmomatic version 0.38.0 was used to remove low quality sequence reads and to trim Illumina adaptors (30). *De novo* assembly was performed using SPAdes version 3.14.0, excluding contigs of <300 bp (31). The PATRIC Similar Genome Finder indicated that MNPAG and MNPAB are most closely related to laboratory strain *P. aeruginosa* PA103 (GenBank accession number JARI00000000) (17). Bowtie2 was used to map trimmed sequencing reads to the sequence of *P. aeruginosa* strain PA103 (22). Annotation was performed using the NCBI Prokaryotic Genome Annotation Pipeline version 4.11 (32). The *Pseudomonas* multilocus sequence type version 2.16.1 (MLST) database categorized both strains as sequence type 446 (ST446) (33).

**Data availability.** This whole-genome shotgun project has been deposited at DDBJ/ENA/GenBank under accession numbers JAAVNI000000000 for *Pseudomonas aeruginosa* MNPAB and JAAVNH000000000 for *Pseudomonas aeruginosa* MNPAG. FASTQ files of raw sequence reads have been deposited under Sequence Read Archive accession numbers SRR11445275 for MNPAB (JAAVNI000000000.1) and SRR11445484 for MNPAG (JAAVNH000000000.1).

## SUPPLEMENTAL MATERIAL

Supplemental material is available online only.

**SUPPLEMENTAL FILE 1**, XLSX file, 0.1 MB.
**SUPPLEMENTAL FILE 2**, XLSX file, 0.03 MB.
**SUPPLEMENTAL FILE 3**, XLSX file, 0.02 MB.
**SUPPLEMENTAL FILE 4**, PDF file, 2.2 MB.

## ACKNOWLEDGMENTS

This work was supported by the National Institutes of Health through grant number R01 AI055042 to T.L.Y.

We thank Patrick Schlievert for the initial observation of and sharing the *Pseudomonas aeruginosa* isolates from the KID patient.

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
