## [Reviewer comments · Microbiology Spectrum]

**Microbiology
Spectrum**

Genome sequences of two *Pseudomonas aeruginosa* isolates with defects in type III secretion system gene expression from a chronic ankle wound infection

Sardar Karash, Robert Nordell, Egon Ozer, and Timothy Yahr

Corresponding Author(s): Timothy Yahr, University of Iowa

Review Timeline:

Submission Date:	May 19, 2021
Editorial Decision:	June 11, 2021
Revision Received:	June 18, 2021
Accepted:	June 23, 2021

Editor: Daria Van Tyne

Reviewer(s): The following individuals involved in review of your submission have agreed to reveal their identity: Thomas Wood (Reviewer #2)

Transaction Report:

DOI: <https://doi.org/10.1128/Spectrum.00340-21>

June 11, 2021

Dr. Timothy L Yahr
University of Iowa
Dept. of Microbiology
540B EMRB
Iowa City, Iowa 52242

Re: Spectrum00340-21 (Genome sequences of two *Pseudomonas aeruginosa* isolates with defects in type III secretion system gene expression from a chronic ankle wound infection)

Dear Dr. Timothy L Yahr:

Thank you for submitting your manuscript to Microbiology Spectrum. Your manuscript was reviewed by two expert reviewers. Both the reviewers and I support publication of a revised paper. Please revise the paper along the lines suggested by the reviewers, making sure that you address all reviewer comments.

I agree with both reviewers that the WGS data analysis presented in your study needs to be developed further. In this regard I have two comments:

1. You note that MNPAB and MNPAG both belong to the same ST, thus I would like to know how closely related they are to one another. Their genome lengths differ by >350Kb, suggesting there might be differences in mobile genetic element content (maybe a mega-plasmid, or perhaps multiple prophages or ICEs) between them. These should be investigated and described.

2. I agree with the reviewers that the use of PA103 as a reference genome needs to be further motivated/explained, or perhaps a different, more closely related reference sequence should be used to identify mutations in MNPAB and MNPAG. I also would like to know what genetic differences (i.e. mutations) are identified when MNPAB and MNPAG are compared directly to one another, and if any of these differences could explain the observed difference in PexsD-lacZ reporter activity, secreted ExoU, and ExsA expression in the strains upon CyaB expression.

When submitting the revised version of your paper, please provide (1) point-by-point responses to the issues raised by the reviewers as file type "Response to Reviewers," not in your cover letter, and (2) a PDF file that indicates the changes from the original submission (by highlighting or underlining the changes) as file type "Marked Up Manuscript - For Review Only". Please use this link to submit your revised manuscript - we strongly recommend that you submit your paper within the next 60 days or reach out to me. Detailed information on submitting your revised paper are below.

Link Not Available

Sincerely,

Daria Van Tyne

Journals Department
Reviewer comments:

Reviewer #1 (Comments for the Author):

Summary:

In this study, Karash and colleagues examined the loss of type 3 secretion system (T3SS) activity in two clinical isolates of *P. aeruginosa* taken from a chronic ankle wound in a person with the genetic disorder keratitis-ichthyosis-deafness (KID) syndrome. The authors used a combination of comparative genomics and targeted in vitro assays to identify the mechanism of T3SS in these isolates. Providing ExsA in trans restored T3SS activity in both isolates and, looking further upstream in the regulatory cascade, providing CyaB in trans to restore cAMP production led to T3SS expression in one of the two isolates. This suggests that the loss of T3SS activity is attributable to two different mutations, although the authors were unable to identify the causative mutations by comparing each genome to a reference (PA103) and complementing a few of the likely candidate allelic variants (*fimV* and *bifA*). Overall, the authors conclude that similar to its evolution during chronic infection of the CF airways, the evolution of mutations that inactivate T3SS are important for *P. aeruginosa* persistence in this chronic skin infection that occurred in a person with KID syndrome.

Critique:

This study provides useful and interesting insight into host adaptation of *P. aeruginosa* outside of the context of chronic CF respiratory infections, which is comparatively understudied. The loss of T3SS activity provides some evidence that *P. aeruginosa* may employ a common route of evolution toward persistence in a human host environment, whether it is in CF (reviewed in PMID: 26946977, non-CF bronchiectasis (e.g. PMID: 28446558), or in KID syndrome. Aside from a couple of technical suggestions regarding the use of a reference genome versus de novo assemblies (described in the "Major comments" section), my main critique of this paper is that it stops short of identifying the causative mutations leading to T3SS inactivation in these two isolates. However, considering the comparative gap in the literature regarding how *P. aeruginosa* evolves to persist in human hosts outside of the context of CF, I think it is informative to describe the isolates from KID syndrome and their genomes will be valuable to researchers broadly interested in *P. aeruginosa* evolution.

Major comments:

(1) Issues related to use of a reference genome (PA103)

1a) It is not clear to me how distant the two clinical isolates are from the reference genome that they used to map their reads to and call mutations (PA103). It would be helpful if the authors included a phylogenetic tree based on core genome SNPs (e.g. using roary) with each clinical isolate, PA103, and a couple of other *P. aeruginosa* reference strains from each of the main clades (e.g. PAO1, PA14, and PA7; clades described in Figure 1A of PMID: 26483767).

1b) By mapping reads to a reference genome (PA103), the authors may miss mutations occurring in genes that are not present in the reference genome. If the two clinical isolates are more closely related to each other than to PA103, then the authors should use one of the de novo assembled clinical isolate genomes as a reference genome and map reads from the other clinical isolate to this, in order to identify variants that may explain the loss of T3SS activity. breseq is an easy-to-use tool for this type of analysis (PMID: 24838886).

Minor comments:

(2) The authors should provide a supplemental table with the mutations they identified in these two strains.

(3) The authors describe an experiment providing BifA in trans (lines 153-154), but say "data not shown". The authors should include a figure with this data.

(4) Line 180: The carrot sign is backwards-the authors probably excluded contigs < 300 bp, not > 300 bp.

Reviewer #2 (Comments for the Author):

This study by Karash et al. describes two *Pseudomonas aeruginosa* strains, isolated from a chronic cutaneous wound from a keratitis-ichthyosis-deafness (KID) syndrome patient, in terms of their type III secretion system (T3SS) activity. Through assessment of an exsD transcriptional reporter and secretion of T3SS substrates, the authors show that both isolates are deficient in expression and activation of the T3SS, but that provision of the transcriptional activator ExsA in trans rescues this defect. Upstream in the regulatory cascade for T3SS activity, ectopic production of the cAMP-dependent activator Vfr has little effect; however, CyaB expression restores T3SS expression and activity in one isolate, MNPAB. Nevertheless, CyaB has no impact on T3SS activity in the other isolate MNPAG. The authors conclude that the loss of type III secretion in these non-CF clinical isolates is through distinct mechanisms, proposing that T3SS deficiency may be a more frequent occurrence in chronic non-CF *P. aeruginosa* isolates than previously appreciated. The study is straightforward, with claims supported adequately by the data presented. This work, in tandem with the corresponding deposited sequences, sets the groundwork for furthering our understanding of the regulatory cascades governing the T3SS in *P. aeruginosa* in the context of human infection.

Minor comments:

1) Please describe or reference the plasmids used to ectopically express exsA, vfr and cyaB. Are the genes under the control of a constitutive promoter?

2) Please add marker sizes for the immunoblots and Coomassie-stained gels

3) The description of the genome sequences could be more informative with a figure and perhaps some expansion. PA103 is claimed to be a "related" strain; however, the large number of polymorphisms precludes identification for the factor(s) responsible for the T3SS defects. Would

the large number of polymorphisms prevent inclusion of a table documenting those of interest?
4) A comment on the major differences, for example in virulence factor genes, between the isolates and PA103 would be informative, since the differences in predicted CDS number appears to amount to hundreds.

Staff Comments:

Preparing Revision Guidelines

For complete guidelines on revision requirements, please see the Instructions to Authors at [link to page]. **Submissions of a paper that does not conform to Microbiology Spectrum guidelines will delay acceptance of your manuscript.**

Please return the manuscript within 60 days; if you cannot complete the modification within this time period, please contact me. If you do not wish to modify the manuscript and prefer to submit it to another journal, please notify me of your decision immediately so that the manuscript may be formally withdrawn from consideration by Microbiology Spectrum.

If you would like to submit an image for consideration as the Featured Image for an issue, please contact Spectrum staff.

Spectrum00340-21 Review

Comments for the authors

This study by Karash *et al.* describes two *Pseudomonas aeruginosa* strains, isolated from a chronic cutaneous wound from a keratitis-ichthyosis-deafness (KID) syndrome patient, in terms of their type III secretion system (T3SS) activity. Through assessment of an *exsD* transcriptional reporter and secretion of T3SS substrates, the authors show that both isolates are deficient in expression and activation of the T3SS, but that provision of the transcriptional activator ExsA *in trans* rescues this defect. Upstream in the regulatory cascade for T3SS activity, ectopic production of the cAMP-dependent activator Vfr has little effect; however, CyaB expression restores T3SS expression and activity in one isolate, MNPAB. Nevertheless, CyaB has no impact on T3SS activity in the other isolate MNPAG. The authors conclude that the loss of type III secretion in these non-CF clinical isolates is through distinct mechanisms, proposing that T3SS deficiency may be a more frequent occurrence in chronic non-CF *P. aeruginosa* isolates than previously appreciated. The study is straightforward, with claims supported adequately by the data presented. This work, in tandem with the corresponding deposited sequences, sets the groundwork for furthering our understanding of the regulatory cascades governing the T3SS in *P. aeruginosa* in the context of human infection.

Minor comments:

- 1) Please describe or reference the plasmids used to ectopically express *exsA*, *vfr* and *cyaB*. Are the genes under the control of a constitutive promoter?
- 2) Please add marker sizes for the immunoblots and Coomassie-stained gels
- 3) The description of the genome sequences could be more informative with a figure and perhaps some expansion. PA103 is claimed to be a “related” strain; however, the large number of polymorphisms precludes identification for the factor(s) responsible for the T3SS defects. Would the large number of polymorphisms prevent inclusion of a table documenting those of interest?
- 4) A comment on the major differences, for example in virulence factor genes, between the isolates and PA103 would be informative, since the differences in predicted CDS number appears to amount to hundreds.

RESPONSE TO EDITOR AND REVIEWER COMMENTS

Editor comment 1: You note that MNPAB and MNPAG both belong to the same ST, thus I would like to know how closely related they are to one another. Their genome lengths differ by >350Kb, suggesting there might be differences in mobile genetic element content (maybe a mega-plasmid, or perhaps multiple prophages or ICEs) between them. These should be investigated and described.

Author response: We now include a phylogenetic analysis as Fig. S1 comparing MNPAB and MNPAG to each other and the common laboratory reference strains PA103, PA14, PAK, PAO1, and PA7 (also suggested by reviewer 1). That analysis shows MNPAB and MNPAG to be highly related, with strain PA103 being the closest match to a reference strain. The latter finding justifies our decision to use strain PA103 as the reference genome for mapping reads. We also searched the whole genome and the 350 kb, no plasmids, prophages, mobile elements were identified.

Editor comment 2: I agree with the reviewers that the use of PA103 as a reference genome needs to be further motivated/explained, or perhaps a different, more closely related reference sequence should be used to identify mutations in MNPAB and MNPAG. I also would like to know what genetic differences (i.e. mutations) are identified when MNPAB and MNPAG are compared directly to one another, and if any of these differences could explain the observed difference in PexsD-lacZ reporter activity, secreted ExoU, and ExsA expression in the strains upon CyaB expression.

Author response: As described above, the newly included phylogenetic analysis supports the use of strain PA103 as the reference genome. For our original analyses the MNPAB and MNPAG genomes were de novo assembled and compared to each other, and to strain PA103. We now include 3 supplemental tables that provide each of these comparisons. As stated in the original submission there are many SNPs that might account for the loss of T3SS gene expression in both MNPAB and MNPAG.

Reviewer 1 Comment: Issues related to use of a reference genome (PA103)

1a) It is not clear to me how distant the two clinical isolates are from the reference genome that they used to map their reads to and call mutations (PA103). It would be helpful if the authors included a phylogenetic tree based on core genome SNPs (e.g. using roary) with each clinical isolate, PA103, and a couple of other *P. aeruginosa* reference strains from each of the main clades (e.g. PAO1, PA14, and PA7; clades described in Figure 1A of PMID: 26483767).

Author response: This has been addressed as described above with Fig. S1. We used 2 different tools to find the closest laboratory strains for MNPAB and MNPAG. Mash: MinHash PMID: 27323842, and PGFams with RAxML. Both showed PA103 is the closest to our isolates. We only presented results from the latter analysis.

Reviewer 1 Comment: By mapping reads to a reference genome (PA103), the authors may miss mutations occurring in genes that are not present in the reference genome. If the two clinical isolates are more closely related to each other than to PA103, then the authors should

use one of the de novo assembled clinical isolate genomes as a reference genome and map reads from the other clinical isolate to this, in order to identify variants that may explain the loss of T3SS activity. breseq is an easy-to-use tool for this type of analysis (PMID: 24838886).

Author response: This is an important point that was part of our original analyses. The MNPAB and MNPAG genomes were de novo assembled and compared to each other, and to strain PA103. We now include 3 supplemental tables that provide each of these comparisons (Tables S1-3).

Reviewer 1 Comment: The authors should provide a supplemental table with the mutations they identified in these two strains.

Author response: As mentioned above, we now include supplemental tables that list the mutations identified in MNPAB and MNPAG.

Reviewer 1 Comment: The authors describe an experiment providing BifA in trans (lines 153-154), but say "data not shown". The authors should include a figure with this data.

Author response:
We now present the results in Figure S3.

Reviewer 1 Comment: Line 180: The carrot sign is backwards-the authors probably excluded contigs < 300 bp, not > 300 bp.

Author response: Corrected

Reviewer 2 Comment: Please describe or reference the plasmids used to ectopically express *exsA*, *vfr* and *cyaB*. Are the genes under the control of a constitutive promoter?

Author response: The *exsA*, *vfr*, and *cyaB* expression plasmids were described previously. Each gene is under the transcriptional control of the arabinose-inducible pBAD promoter. The methods have been updated to include this information and the appropriate citations.

Reviewer 2 Comment: Please add marker sizes for the immunoblots and Coomassie-stained gels

Author response: Unfortunately, that information is not available for those experiments, short of repeating them, which seems unnecessary. The Coomassie stained pattern is very distinctive and not open to interpretation with >25 years of experience in performing that particular experiment. Similarly, the specificity of the immunoblots is well-established.

Reviewer 2 Comment: The description of the genome sequences could be more informative with a figure and perhaps some expansion. PA103 is claimed to be a "related" strain; however, the large number of polymorphisms precludes identification for the factor(s) responsible for the T3SS defects. Would the large number of polymorphisms prevent inclusion of a table

documenting those of interest?

Author response:

As mentioned above, this has been addressed by addition of Fig. S1 and Tables S1-3.

Reviewer 2 Comment: A comment on the major differences, for example in virulence factor genes, between the isolates and PA103 would be informative, since the differences in predicted CDS number appears to amount to hundreds.

Author response: The only notable difference is the absence of the hcn gene cluster in MNPAB, which is responsible for hydrogen cyanide production.

June 23, 2021

Dr. Timothy L Yahr
University of Iowa
Dept. of Microbiology
540B EMRB
Iowa City, Iowa 52242

Re: Spectrum00340-21R1 (Genome sequences of two *Pseudomonas aeruginosa* isolates with defects in type III secretion system gene expression from a chronic ankle wound infection)

Dear Dr. Timothy L Yahr:

Thank you for submitting a revised version of your manuscript. Your manuscript has now been accepted, and I am forwarding it to the ASM Journals Department for publication. You will be notified when your proofs are ready to be viewed.

Sincerely,

Daria Van Tyne
Editor, Microbiology Spectrum

Journals Department
Table S1: Accept
Fig S1-3: Accept

Table S3: Accept
Table S2: Accept